# Stabilisation of Viral Membrane Fusion Proteins in Prefusion Conformation by Structure-Based Design for Structure Determination and Vaccine Development

**DOI:** 10.3390/v14081816

**Published:** 2022-08-18

**Authors:** Henriette Ebel, Tim Benecke, Benjamin Vollmer

**Affiliations:** 1Centre for Structural Systems Biology (CSSB), 22607 Hamburg, Germany; 2Department of Chemistry, University of Hamburg, 20146 Hamburg, Germany; 3Leibniz Institute of Virology (LIV), 20251 Hamburg, Germany

**Keywords:** viral fusion protein, membrane fusion, fusion mechanism, prefusion, postfusion, protein stabilisation, structure-based design, glycoprotein B, fusogen, herpesvirus

## Abstract

The membrane surface of enveloped viruses contains dedicated proteins enabling the fusion of the viral with the host cell membrane. Working with these proteins is almost always challenging because they are membrane-embedded and naturally metastable. Fortunately, based on a range of different examples, researchers now have several possibilities to tame membrane fusion proteins, making them amenable for structure determination and immunogen generation. This review describes the structural and functional similarities of the different membrane fusion proteins and ways to exploit these features to stabilise them by targeted mutational approaches. The recent determination of two herpesvirus membrane fusion proteins in prefusion conformation holds the potential to apply similar methods to this group of viral fusogens. In addition to a better understanding of the herpesviral fusion mechanism, the structural insights gained will help to find ways to further stabilise these proteins using the methods described to obtain stable immunogens that will form the basis for the development of the next generation of vaccines and antiviral drugs.

## 1. Introduction

Many important human pathogens such as the large family of herpesviruses as well as the majority of emerging disease viruses are enveloped viruses. In order to overcome the membrane barrier, set by the host cell, a variety of different proteins evolved to efficiently fuse the viral with the cellular membrane. During this process, these viral membrane fusion proteins transition from a metastable prefusion conformation like a loaded spring present on the virus particle, to the lowest-energy postfusion or hairpin-like form. Usually, this change is irreversible, leaving the postfusion proteins on the host cell membrane after entry. The final merging of the two membranes releases the intraviral content, including the viral genome into the cytoplasm of the host cell where the replication cycle can be initiated. Since viral membrane fusion proteins are a prime target of the immune system, they are of great medical relevance. Unfortunately, the structure determination of these proteins, especially in prefusion conformation, is often difficult due to the inherent metastability, but several strategies have been developed to overcome these hurdles. Ultimately, the gained structural information holds the potential to generate the next generation of vaccines and antiviral drugs to counteract existing and new viral threats.

## 2. The Fusion Process

There are different energy barriers preventing the spontaneous fusion of biological membranes in the natural environment of a cell. One is set by the hydration force, a strong short-range repulsive force occurring between two polar membrane surfaces separated by a thin (20–30 Å) layer of water. To initiate membrane fusion, the water molecules held by the electrostatic force of the hydration shell of the membrane must be removed first [1]. After the merging of the outer membrane leaflets, the fusion process can only conclude by overcoming the second energy barrier set by hydrophobic interactions of the lipids that prevent the spontaneous rupture of the membrane.

The viral membrane fusion proteins embedded in the viral envelope are the key to catalyse this membrane fusion reaction that opens the way to release the pathogen’s genome into the cytoplasm of the host [2]. Host cell receptor binding directs these proteins to the position where they are able to fulfil their function. Specific triggers like a pH shift in the endosome or the receptor attachment itself, start off a cascade of defined conformational changes leading to an extension towards the host cell membrane and insertion of previously shielded hydrophobic fusion peptides or loops [3] (Figure 1, Insertion). Once hooked, the extended state collapses and the protein folds back onto itself, pulling the two membranes into close proximity, likely in the form of protrusions to limit repulsive forces [4] (Figure 1, Collapse). This allows the formation of a hemifusion stalk which radially extends to a hemifusion diaphragm where lipids of the outer membrane layers start mixing while the inner membrane leaflets are still separate (Figure 1, Backfold). Probably due to the local membrane tension induced by the fusion proteins, the hemifusion diaphragm ruptures and opens a fusion pore, completing the full merger of the two membranes [4,5,6,7]. In the end, the fusogen is left in a stable postfusion conformation (Figure 1, Postfusion).

Since the first viral membrane fusion protein structure was determined for influenza haemagglutinin (HA) [8], many more followed from a variety of different viruses. Structural elements found in all of these proteins are the lipophile fusion peptides or loops that need to be shielded from the environment before target membrane insertion and membrane anchors for localisation and attachment on the surface of the viral particle. Although there are significant structural differences, the identification of common structural elements permits the assignment of most of these proteins to one of three classes.

## 3. The Classes of Viral Membrane Fusion Proteins

### 3.1. Class I

Influenza HA is the defining protein for the group of class I fusion proteins [9]. The trimeric structure shows features that are found in fusion proteins of other unrelated viruses, including a central triple stranded coiled-coil region and the formation of an antiparallel 6-helix bundle in the postfusion hairpin conformation [10]. The N-terminal fusion loops are linked to a short α-helix that is connected via a long loop region to the central, long α-helix that forms the coiled-coil in the trimer (Figure 1A Prefusion). Although the structures of the pre- and postfusion conformation are well characterised, only recently were additional intermediate states of the fusion process structurally described [11]. After triggering the fusion process, the connecting loop region is released by protease cleavage and refolds to become part of the central helix, thereby extending the structure to bridge the gap towards the host membrane where the fusion loops, now located at the tip, can be inserted (Figure 1A Insertion). The formation of the characteristic hairpin conformation pulls the two membranes together by refolding a C-terminal portion of the central α-helix to create the antiparallel 6-helix bundle (Figure 1A, Postfusion).

Fusogens with similar structural elements were found in a variety of different viruses including the human immunodeficiency virus (HIV) glycoprotein 41 (gp41) [12], respiratory syncytial virus (RSV) F1 [13] or the ebolavirus glycoprotein GP [14]. Class I fusogens consistently rely on receptor binding as part of their fusion initiation. Here, the species of bound receptors varies between different viruses and interactions often fulfil different functions including attachment, internalisation or fusion triggering [15]. For example, influenza HA binds to sialic acids in the host cell glycocalyx, which coordinates transport to the acidified endosome for pH activation [16] while Ebola GP1 needs cleavage by low-pH dependent cellular proteases to present a binding pocket specific for the Niemann–Pick C1 receptor [17]. In the case of RSV, several receptor candidates have been proposed, facilitating entry by interaction with the RSV fusion protein, but the exact details of the mechanism are still unclear [18]. The comparative analysis of the members of this class revealed the precursor protein cleavage as another common functional feature, and often a prerequisite for activation. The processed proteins are arranged in heterodimers forming one protomer each which in turn assemble into homotrimers [2].

### 3.2. Class II

The earliest structures determined for viral fusion proteins that were subsequently grouped in class II included protein E of flaviviruses [19] and E1 of alphaviruses [20] which belong to the family of Togaviridae. The first examples of the two viral families revealed very similar structures, mainly composed of β-sheets and arranged in three connected domains. Both proteins are expressed as polyproteins that also include a chaperon protein (prM for flaviviruses and pE2 for alphaviruses) from which they are cleaved post-translationally. The chaperones shield the hydrophobic fusion peptides of E and E1 during viral maturation prior to the arrangement of primed homodimers for flavivirus E and homotrimers of E1:E2 for alphaviruses on the viral membrane [2]. In contrast to class I, the orientation of the proteins is parallel to the membrane and the superstructural arrangement showing icosahedral symmetry follows that of the capsid [21]. Although the general steps of the fusion process are similar to class I, meaning expansion of the proteins towards the host membrane, fusion loop embedding and membrane approximation followed by merging, the details of the structural rearrangements in class II proteins are quite different (Figure 1B). The extension to present the fusion peptides does not include secondary structure refolding like the class I α-helix extension, but is achieved by the rearrangement of the three domains. The initiation of the fusion process is achieved in all known class II viral fusion proteins by the low pH shift in the acidic environment of the endosomal compartments [22] leading to protonation of mainly histidines, which results in the following conformational changes [23]. The rotation of domain III (Figure 1B, blue box) around the connection to domain I (Figure 1B, green box) first exposes the fusion loop located at the distal end of domain II (Figure 1B, light orange box) to the target membrane. Once embedded in the membrane, the protein pulls back, forming a hairpin-like structure with domains I and II packed against domain III and the region that forms the connection to the transmembrane helix [24,25]. The postfusion structures are both homotrimers and therefore require the flavivirus E protein dimers and the alphavirus E1:E2 complex to dissociate and rearrange between activation and hairpin formation [25,26]. Similar to the helical hairpin-like arrangement of class I fusion proteins, the fusion loops end up close to the transmembrane helices (Figure 1B, Postfusion).

### 3.3. Class III

When the structures of herpes simplex virus 1 (HSV-1) glycoprotein B (gB) [27] and the glycoprotein G (G) of Vesicular Stomatatis virus (VSV) [28,29] were revealed, they showed structural features of the first two classes, including the central α-helical coiled-coil and internal fusion loops located at the tip of a domain composed of β-sheets [30] (Figure 1C). The surprising but close structural relationship between the two proteins, and the little resemblance to any of the canonical class I or II proteins, prompted the definition of a third class [31]. In addition to more fusion proteins from other herpesviruses, namely pseudorabies virus (PrV) gB [32], varicella zoster virus (VZV) gB [33], human cytomegalovirus (HCMV) gB [34] and Epstein–Barr virus (EBV) gB [35] and G protein of rabies virus [36,37], gp64 from baculovirus [38] and the Thogoto and Dhori virus Gps [39] were added to the class due to their structural homology with the postfusion structures of HSV-1 gB and VSV G.

The ectodomain of gB can be divided into five subdomains of which all but domain V have structural equivalents in VSV G [40]. Domain I (Figure 1C, light orange box), called fusion domain in VSV G, is mainly formed by β-sheets and contains internal fusion loops at its tip as found in class II fusion proteins [30]. The recently solved structure for the prefusion conformation of HCMV gB shows how these fusion loops are shielded by the membrane proximal region (MPR) from the surrounding membrane [41]. Interestingly, so far, no equivalent region could be structurally determined in VSV G, but the position of the ectodomain on the virus determined by electron cryo tomography (cryoET) and subvolume averaging suggests the fusion loops to be located in the viral membrane [42]. Domain II, termed PH domain in VSV G due to the similarity to the pleckstrin-homology superfold, forms the connection to domain III or central domain in VSV G (Figure 1C, purple cylinder). The latter domain is mainly composed of a central α-helix forming a trimeric coiled-coil similar to class I and generates the trimerisation contacts to build up the rod-like shaped trimer of the postfusion conformation. Domain IV in gB, which is considered part of the central domain in VSV G, has no structural relatives and forms the connection to domain V or the unstructured stem in G while both lead to the MPR, followed by the transmembrane helices.

The transition towards the postfusion conformation occurs upon fusion triggering after target cell surface receptor binding for gB or with an additional change in pH for VSV G, but in contrast to class I without precursor cleavage [10]. The fusion trigger then leads to the flipping of the first two domains, thereby releasing and exposing the fusion loops toward the opposite side into close proximity to the target membrane. To do so, in VSV G, parts of the central domain undergo refolding from two shorter helices, separated by a hinge to one extended central helix, as seen in class I fusion proteins (Figure 1C, Insertion). A similar mechanism was proposed for the structural transition in gB [43], but the high-resolution structure of HCMV gB in prefusion conformation showed the central helix in domain III fully intact without a comparable hinge region [41]. While in VSV G all other domains retain their fold [28,29,44], domain V in gB undergoes substantial rearrangement [27,30,41]. Three α-helical parts transform into extended coils that insert between the neighbouring protomers, thereby forming an extensive trimerisation interface. In addition, the C-terminal helical bundle preceding the MPR changes from a closer packing in the prefusion conformation to a wider, funnel-like arrangement in the postfusion form. The refolding of domain V drags the connected membrane embedded regions to the same end of the protein as the fusion loops, pulling the two membranes together (Figure 1C, Collapse—Postfusion). During the conformational changes of gB, a simultaneous domain rotation around the central trimer axis occurs, twisting the protomers around each other [41,43].

## 4. Medical Relevance of the Viral Membrane Fusion Proteins

The essential function of the fusion proteins and their exposure on the viral particle make them prime targets for the immune system and important vaccine antigens, demonstrated by the generation of neutralising antibodies directed against these proteins [14,45]. However, due to their extensive conformational rearrangements during the fusion process, some epitopes are only accessible in either the pre- or postfusion state. The fact that the prefusion form is found on the infectious particle makes it medically more relevant. For the RSV protein F, the prefusion conformation exhibits epitopes that are the target of the most potent neutralizing antibodies, suggesting that subunit vaccines should be based on this form [46]. However, the inherent metastability of these proteins and especially of the prefusion conformation can lead to low expression rates, misfolding, protein instability and aggregation [47,48] or pre-mature triggering into the postfusion state [49,50]. These problems have severely hampered the structural determination and subsequent use for drug and vaccine development of a number of different fusion proteins of human pathogenic viruses. To circumvent the aforementioned problems the membrane anchors can be replaced by trimerisation domains, and the proteins stabilised through introduction of specific functional mutations. Recent years have seen a number of cases where membrane fusion proteins were successfully stabilised, including the HA protein from influenza virus, the F protein from RSV and the spike protein S from severe acute respiratory syndrome coronavirus type 2 (SARS-CoV-2) (see Table 1). The stabilised construct of the latter one is now successfully in use in the form of a subunit and different mRNA vaccines [51]. Nevertheless, the fact that immunisation with these vaccines provides reliable protection by the generation of neutralising antibodies, but their titres decrease significantly several months after vaccination [52], calls for further developments. Similar problems were encountered, among others, during the development of a vaccine using stabilised versions of the HIV membrane fusion glycoprotein [53]. This protein is translated as the precursor gp160, which is activated after glycosylation in the golgi by furin or furin-like protease cleavage into gp120 and gp41 [54]. The final, mature protein is formed by a trimer of non-covalently associated gp120-gp41 heterodimers. Initial trials using a recombinant, monomeric form of gp120 as subunit vaccine did not elicit broadly neutralising antibodies and even triggered the formation of immunodominant non-neutralising antibodies [55]. Over the years, a number of different constructs comprising gp120 and gp41 that contained several modifications in order to better mimic the native trimer of the HIV spike and to stabilise the prefusion form [56] were developed. Now, there is a much better understanding of the weaknesses of the previously used construct designs and their ability to induce broad neutralising antibody responses [56]. In addition to the vast diversity of different HIV strains, which continues to challenge the generation of an effective vaccine, it should be noted that the stabilising modifications can also lead to further complications. The CD4 receptor binding site is the target for some of the most potent broadly neutralising antibodies and might be essential in developing an effective vaccine. The highly stabilised trimer constructs represent a closed form of the spike that lost its metastability, which means the CD4 receptor binding sites stay tucked in between the protomers. This might lead to steric hindrances and be the reason why these constructs often fail to elicit an efficient neutralising antibody response against this important epitope [56].

Taken together, viral membrane fusion proteins represent an excellent, and sometimes only target for antiviral and vaccine development approaches. Prefusion stabilised versions of these proteins are a powerful tool in this regard, as successfully shown for RSV and SARS-CoV-2. Nevertheless, the case of HIV illustrates the possible problems and drawbacks researchers may face in developing effective vaccines with this approach.

## 5. The Herpesviral Fusion System

The fusion machinery of herpesviruses consists of at least three individual surface glycoproteins, encoded in individual genes, to allow for host cell entry [86]. These proteins termed glycoprotein H, L (gH/L) and B (gB) are conserved within the whole herpesvirus family [87]. The bona fide membrane fusion protein in all herpesviruses is the structurally conserved glycoprotein B. The role of the heterodimeric gH/L complex is less clear, but it is supposed to regulate the fusion process by interaction with gB. The herpesviral entry mechanism as well as the proteins involved in this process are described in detail in recent reviews [86,88]. In the prototypic HSV-1, the set of proteins required for cell entry also includes the receptor binding glycoprotein D (gD). Interactions of gD were identified with three different types of cell receptors [89]: the herpesvirus entry mediator which belongs to the tumor necrosis factor receptor superfamily, three members of the Ig-like receptor family, namely nectin-1 and -2 and the poliovirus receptor related protein as well as 3-O-sulfonated-heparan sulfate. Additional connections with the host cell are established by gB via the paired immunoglobulin-like type 2 receptor α [90] and heparan sulfate [91] which is shared with glycoprotein C (gC) [92], but seems to be at least partially dispensable [93]. The ability to interact with this variety of cellular surface molecules might explain the broad cell tropism that includes different neuronal and epithelial cells [89,94]. Depending on the cell type, HSV can enter either by fusion at the plasma membrane or after endocytosis by fusion with intracellular membranes of the endosomal pathway [95]. The latter is supposed to be triggered by low pH leading to conformational changes in gB [96]. Although the postfusion form of gB undergoes minor structural rearrangements when exposed to acidic pH [97], similar direct structural evidence for low pH triggering of the prefusion form of gB to perform the large conformational changes necessary for the membrane fusion process has yet to be provided. Interestingly, the latter process was shown to be regulated by gC, which seems to increase the pH threshold allowing an earlier exit from endosomal compartments [98].

The use of multiple proteins during entry distributes the requirements and tasks of the fusion process including receptor binding, attachment and membrane fusion as well as regulation to different players [99]. gB is a homotrimeric, single-pass, type I transmembrane protein. After the successful determination of the HSV-1 gB structure by X-ray crystallography, several homologues from different herpesvirus subfamilies followed, namely gB of PrV, HCMV and EBV [27,32,34,35]. The resulting structures all show a strong resemblance and were found to all be in the postfusion conformation with the class I typical hairpin-like arrangement. The constructs used in these experiments only comprised the ectodomains of gB and omitted the transmembrane and cytoplasmic domains. The metastability of the protein coupled with the missing stabilising effect of the membrane embedded and attached domains seems to push the ectodomain into postfusion conformation [50]. Similar effects were observed for the parainfluenza virus 3 fusion protein [49], the baculovirus gp64 [38] or the Thogoto and Dhori virus Gps [39].

## 6. Approaches to Stabilise Membrane Fusion Proteins

In recent years, a number of viral membrane fusion proteins have been successfully stabilised in order to improve usability for various applications and also to obtain their prefusion structure (Table 1). Early examples of successful structure stabilisation via introduction of mutations relied on previously solved structures [8,58,100]. This approach was further extended by the use of structural information of close homologues to identify suitable residues to mutate [57,59]. By this, stabilising mutations might make the determination of a first structure possible [57]. The resulting structural information can then in turn be used to produce even more stable constructs suitable for various in vitro applications or immunogen generation [61]. In case no prior structural information is available and the wildtype protein metastability prevents structural determination, stabilisation might be achieved either by the addition of agents such as stabilising antibodies, inhibitors or crosslinkers or by mutation. The later approach of designing stabilising mutations to solve a structure often requires mutational screens and suitable test systems to validate the effect of these mutations. Structural biologists have an extensive toolbox of different approaches they can choose from to stabilise these proteins, including disulfide-bond linkage, reduction of charge repulsion and cavity filling or substitution of residues for prolines. In addition, cleavage site modifications are often introduced and trimerisation domains are employed for structural stabilisation if the fusion protein is purified without its transmembrane domain (Figure 2, Table 1).

### 6.1. Multimerisation Domains

The expression of constructs comprising only the ectodomain of the investigated fusion protein can increase the expression rate and prevents the use of detergent and lengthy purification and, if need be, reconstitution methods. However, in some cases, the interaction between the ectodomain protomers is too weak to maintain the trimeric conformation without the membrane anchor causing the protein complex to fall apart into monomers [101]. To achieve the stabilisation of these proteins in a trimer, an artificial multimerization domain such as the T4 fibritin trimerisation motif (foldon) or the modified GCN4-pII isoleucine zipper domain can be introduced to fulfil this function. The foldon domain consists of three copies of the fibritin C-term, comprising 30 amino acids and forming a β-propeller-like structure with strong interactions between the subunits [102,103] enforcing the trimerisation and also correct alignment of the associated fusion protein ectodomains (Figure 2A) [57,59,61,67,81]. In comparison, the modified yeast transcriptional activator GCN4-pII isoleucine zipper domain assembles from three 33 amino acid long α-helices, forming a parallel three-stranded, coiled-coil [104] and thereby promoting the trimeric arrangement of the attached ectodomains [64,79,83,84].

**Figure 2 viruses-14-01816-f002:**
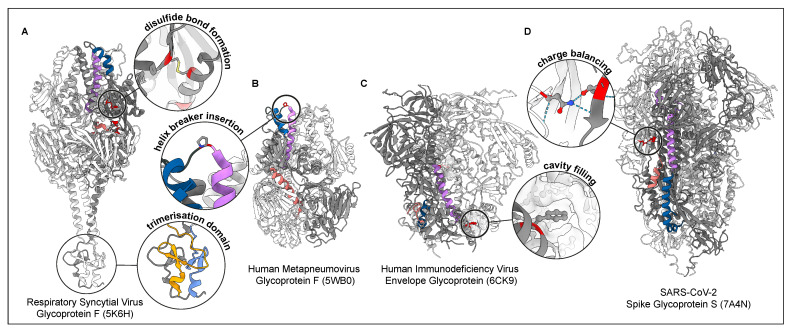
Examples of stabilising modifications. Viral membrane fusion protein trimers stabilised in prefusion conformation by different mutations (red) depicted in ribbon representation [105] with insets showing specific examples of each stabilisation in detail. One protomer of each trimer is emphasised (dark grey) with structural elements essential for the fusion process coloured according to Figure 1: fusion peptides (light red), central helix (purple) and short helix (blue). PDB accession numbers are given in parenthesis for each structure. (**A**) Respiratory Syncytial Virus F glycoprotein [67] stabilised through interprotomer, intramolecular disulfide bond formation (yellow) and a T4 fibritin trimerisation domain (with protomers coloured grey, orange and light blue). (**B**) Human metapneumovirus F glycoprotein [57] stabilised by residue substitution to proline located at the hinge region between the short and central helix acting as helix breaker to prevent helix extension. (**C**) Human immunodeficiency virus envelope glycoprotein [77] stabilised through cavity filling mutations showing a phenylalanine inserted into the protein cavity of a neighbouring protomer (white surface). (**D**) Severe acute respiratory syndrome coronavirus type 2 (SARS-CoV-2) spike protein [63] stabilised by charge balancing, whereby a repulsive force is replaced by an attractive force by replacing aspartate by asparagine to enable hydrogen bonding (dotted lines between red (oxygen) and blue (nitrogen) coloured atoms).

In some cases, the addition of one of these artificial trimerisation domains to the ectodomain construct is enough to stabilise the fusion protein in a prefusion conformation, as shown for protein F of parainfluenza virus 5 [101]. Although a viable tool for structure determination, it should be noted that multimerization domains can elicit potent antibody responses when used as part of protein or subunit vaccines. In case of GCN4, this effect can be mitigated by the introduction of four potential N-linked glycosylation sites [106].

### 6.2. Disulfide Bonds

The cysteine-linkage strategy based on the introduction of disulfide bonds is assumed to lower the entropy of the intermediate states and thereby increase the conformational stability of the prefusion structure [107]. There are different ways to make use of this approach. One is to use disulfides as interprotomer bridges that stabilise the trimeric conformation, such as in gp120/gp41 of HIV [68,76], glycoprotein 1/2 (GP1/GP2) of lassa virus (LASV) [78], RSV F protein [67] (Figure 2A) or influenza HA [69], where the head domains are linked together to prevent their dissociation.

The interprotomer linkage by cysteines has been the first and until now only used approach to stabilise class II fusion proteins. Based on the previously identified dimer stabilisation of dengue virus protein E by antibody binding leading to very stable constructs, potential residues facing themselves in the dimer were tested for residue substitution to cysteine. By this, a successful stabilisation was identified, forming a new disulfide bond in the middle of the dimer interface [70,71]. For further stabilisation, a cysteine double mutant was created, additionally locking the dimer at each end [71]. Based on this, the approach was used as blueprint to generate stabilised Zika virus E protein by using the homologous positions for cysteine substitutions [70,72,73].

Another application is to introduce cysteines to stabilise regions undergoing conformational changes during the fusion process by formation of intramolecular links. In case of Hendra virus protein F, the fusion peptide was anchored to conformationally invariant residues of the head domain to prevent the overall transition from pre- to postfusion conformation [64]. Another possibility is to bridge the head and stalk domains, as done in measles virus protein F [65] or to introduce disulfide bonds into the MPR, as in parainfluenza virus 5 [66]. In general, the introduction of disulfide bonds for structure stabilisation is mostly performed between residues that are adjacent in the prefusion conformation, but come apart during fusion and end up on different sites in the postfusion structure. Although a powerful approach, it either requires detailed structural knowledge about the protein that is to be stabilised or (sometimes extensive) mutation screens that allow the assessment of properly folded proteins and conformational stability by, e.g., cell surface localisation and recognition by conformation-specific antibodies [59].

### 6.3. Helix Breaker Insertion

Residue substitutions for prolines, also termed helix breaker mutations, are particularly well suited for structure-guided stabilisation of class I membrane fusion proteins (for a detailed review, see [108]). The cyclic structure with restricted backbone dihedral angles of prolines disfavours helix formation and confers local rigidity. The helix breaker mutations are introduced in the loop region connecting the central long and short α-helix that refolds to become part of the long central coiled-coil region in the postfusion conformation (Figure 1A and cf. Section 3). The rigidity of the inserted proline residue(s) restricts the structural transition of the hinge region and thereby the formation of the postfusion hairpin conformation. One of the first examples to stabilise the prefusion conformation of a viral fusion protein was achieved for influenza HA based on previously determined structures [8,100]. By the introduction of mutations into the region that forms the coiled-coil helix, the mutated protein was functionally inhibited and showed stabilisation of the prefusion conformation [58].

The metastability, inherent to many membrane fusion proteins in prefusion conformation, can thwart their structural determination. Without prior structural information, it is often difficult to pinpoint suitable positions to introduce stabilising mutations. The proline substitution I559P in HIVgp140, which is a soluble form of the envelope protein that has gp120 stably linked to gp41 by an intermolecular disulfide bond, was the first used proline mediated stabilisation for structure determination. The mutation was identified by empirical testing of various amino acid positions for substitution to proline in the N-terminal heptad repeat region of gp41 to prevent the formation of the 6-helix bundle [76].

In the case of the RSV F protein, the prefusion structure was determined by stabilisation with antibodies [109] and was subsequently used to design stabilised versions by introduction of mutations [47] (Figure 2A). Interestingly, a construct of RSV F stabilised by proline substitution turned out to have the mutation localised in the structurally corresponding position found in HIV gp140 [47]. This surprising finding reveals a conservation of the intricate structure-function relationship, despite the very distant phylogeny of these two viruses.

Similar approaches that proved to be successful for RSV F and HIV gp140 were used to stabilise LASV GP1/GP2 protein [78], human metapneumovirus F protein [79] (Figure 2B) and the Ebola GP protein [75] by introduction of proline substitutions and to subsequently determine their prefusion structures. Furthermore, investigation into stabilising the Middle East respiratory syndrome coronavirus S2 protein to increase expression yields found a beneficial effect from substituting two consecutive residues to prolines in the loop between the short α-helix and the long central helix. The designed mutant, termed 2P, showed a >50-fold increase in expression yield and an overall improved conformational homogeneity [59]. Similar effects were seen for the homologous SARS-CoV-1 and human coronavirus HKU1 spike proteins [60]. To reach even further stabilisation, proline substitutions of up to six residues were introduced to allow and greatly facilitate the expression and structure determination of the prefusion form of the SARS-CoV-2 spike glycoprotein [61]. The resulting construct termed ‘hexapro (6P)’ was subsequently used by countless labs to investigate SARS-CoV-2 directed antibodies [110,111].

Despite the low sequence conservation within class I fusion proteins and often very distant phylogenetic relationships between the harbouring viruses, these proteins share many structural features. The central segment that extends during the fusion process by formation of an extended α-helical coiled-coil is one of the characteristic structural features (cf. Section 3), which makes the hinge loop stabilisation approach generally applicable to class I fusion proteins.

### 6.4. Cavity Filling

Another strategy is to analyse hydrophobic cavities located in regions that differ between pre- and postfusion conformation. By filling these cavities with bulkier amino acid side chains, the free energy is lowered in the prefusion conformation due to a closely packed protein core and increased in the postfusion state by the shielding of hydrophobic residues from solvent [112]. Overall, the cavity filling approach leads to the stabilisation of the prefusion state and was used for further structure stabilisation of Nipah virus (NiV) protein F [79], HIV gp140 [77] (Figure 2C) and RSV protein F [80] for vaccine development. The cavity filling approach requires high resolution structural information of the protein of interest and is therefore suitable for the expression and construct stabilisation rather than initial structure determination [59].

### 6.5. Charge Balancing

Finally, charge modification is helpful to stabilise the protein structure and to prevent electrostatic repulsion between two similarly charged amino acids. Hereby, a charged amino acid is mutated to the opposite charge or to a neutral amino acid which then leads to reduced repulsion, hydrogen bonding and further stabilisation of the protein’s prefusion conformation. As with the cavity filling method, this approach requires high resolution information and is therefore most suited for the later stages of an iterative process that was used to stabilise the prefusion conformation structures of influenza HA [74], SARS-CoV-2 spike S [63] (Figure 2D) and RSV protein F [47].

### 6.6. Combinatorial Mutational Approaches

A range of different viral membrane fusion proteins were successfully stabilised by combining several of the described approaches (Table 1), often on the basis of pre-existing prefusion structures. Examples are the HIV gp140 protein [77], the F proteins of RSV [67] and NiV [79] as well as the SARS-CoV-2 spike S [63] which are also used in vaccine development.

### 6.7. Small Molecule Fusion Inhibitors

There is an ever-growing list of small molecule inhibitors being discovered and developed against various different viral fusion proteins. Their individual interference mechanisms in the fusion process have been covered by a recent comprehensive review [113]. Therefore, only components specifically stabilising the pre-fusion conformation of a small set of example proteins will be addressed. Following the discovery of broadly neutralising antibodies binding the stem of influenza HA, small inhibitors were developed targeting the same functional epitope [114]. These inhibitors act as a molecular ‘glue’ preventing the low pH conformational change that makes fusion with the endosomal membrane possible [115]. In the case of RSV, a group of inhibitors was found to bind a three-fold-symmetric cavity in the prefusion trimer, linking structurally labile regions of the individual protomers and thereby preventing their conformational rearrangements during fusion [116]. One of the extensively studied epitopes recognised by several broadly neutralising antibodies in HIV gp41 is located in the MPR. Although the function of the MPR during the fusion process is still unclear, there are fusion inhibitors targeting a hydrophobic pocket formed by this domain, inhibiting the CD4-induced conformational changes [117]. The recent pre-fusion structure of HCMV gB was determined by the help of a small molecule inhibitor also binding a hydrophobic pocket formed by the MPR and thus locking the bound fusion loop in place, preventing the dissociation required during fusion (cf. Section 7.2) [41].

## 7. Stabilisation of the gB Prefusion Conformation

### 7.1. HSV-1 gB

Despite the addition of trimerisation domains and the introduction of prefusion stabilising as well as postfusion destabilising mutations, efforts to stabilise the soluble ectodomain of HSV gB were unsuccessful in obtaining a prefusion stabilised construct [50]. Although the ectodomain of gB comprises the majority of the protein, the membrane embedded and cytoplasmic domains still amount to a significant portion of the protein, which is not seen in other members of class III and might be related to their important functional roles in gB. For HSV-1 gB, this part of the protein was shown to form an intertwined trimer that is connected to the transmembrane domain adopting an inverted teepee shape [118]. The introduction of specific mutations in the cytoplasmic domain of gB can result in a hyperfusogenic phenotype [119], suggesting a possible stabilising function of this domain. Still, solubilised and purified full-length gB, analysed either by crystallisation in the presence of maltosides and A8-35 amphipol [118] or for VZV gB, by electron cryo microscopy (cryoEM) [33], also adopted the postfusion conformation. Only the purification of the full-length protein, embedded in a native membrane allowed the observation of the prefusion state by electron cryoET for HSV and HCMV gB [42,120]. HSV gB induces the formation of extracellular vesicles upon overexpression in some mammalian cell types. These vesicles are almost exclusively studded with gB proteins [121], permitting the purification of the full-length protein in a native membrane while omitting the need for any solubilisation steps that would remove the natural lipid environment. Interestingly, even with the support of the underlying membrane and the absence of the fusion initiating gH/L complex, about one-third of the vesicles show the protein in postfusion conformation [43], underlining the metastability of the prefusion form. CryoET analysis of the described vesicles followed by subvolume averaging yielded a first structure that outlined the overall shape of the prefusion trimer [120]. The mixture of conformations found on the vesicles causes the deterioration of the achievable resolution as particle picking is difficult and the number of subvolumes that can be used for averaging per tomogram is reduced.

One structural element in VSV G that is also found in class I proteins is the central α-helical region connected by a hinge that refolds during fusion to become part of the extended helix (Figure 1C) [44]. The structural homology between the gB and VSV G postfusion structures suggests a similar transition mechanism for gB. The successful stabilisation of class I fusion proteins by introduction of helix breaker mutations in this region (Table 1) prompted a similar approach for gB [43,50]. The structural alignment of this central domain from different class III fusion proteins revealed specific features, including a conserved leucine or isoleucine residue that precedes the described hinge region in VSV G. The introduction of a helix breaker mutation in the same region in gB rendered the protein incapable of performing fusion while the expression levels remained normal. Interestingly, the introduction of a proline substitution in the corresponding position in VSV-G also permitted expression, suggesting a similarly stabilising effect in fusion proteins of Rhabdoviridae, although a functional inhibition was not tested. The functional stabilisation of gB also proved to be beneficial for structure determination and improved the previously achieved resolution to 9 Å, which allowed the formation of a pseudoatomic ectodomain model [43]. Interestingly, the stabilisation by proline substitution only works in the context of a membrane support, as an ectodomain construct carrying the proline substitution still adopted the postfusion conformation.

### 7.2. HCMV gB

The stabilisation of HCMV gB for structural determination of the prefusion state followed a different approach using a range of ‘externally’ stabilising agents [41]. To do so, purified viral particles were treated with an HCMV gB-specific inhibitor that links the MPR to the embedded fusion loop [122]. In addition, the proteins were chemically fixed using a non-membrane permeable, homobifunctional and amine-reactive crosslinker before binding with a fragment of the neutralising monoclonal antibody SM5-1 [123] that also provided the affinity tag for purification. After the detergent solubilisation out of the viral membrane and the purification the resulting protein particles could be analysed by cryoEM, revealing a small fraction to be in prefusion conformation while the majority were still found to show the postfusion form. It is not clear if part of the proteins adopted the postfusion conformation during the preparation or were already in this form on the viral particles. It might be a combination of both, as postfusion gB was also seen on native HCMV particles by cryoET [42]. The resulting prefusion structure gives important insights into the conformational transitions during the fusion process, highlighting the significant role of domain V in the last steps of fusion. The importance of this domain and its potential influence on the fusion mechanism is also underlined by the fact that specific mutations in this domain in combination with a shortened cytoplasmic domain cause a complete uncoupling of the fusion mechanism from the regulatory influence of gH/L in PrV [124]. Furthermore, the determination of parts of the membrane embedded regions is a significant step towards a better understanding of the role this, in general, structurally underexplored region plays. Comparison of the MPR and transmembrane domain to the corresponding regions in postfusion HSV-1 gB [118] shows only minor changes, including a steeper angle for the transmembrane helices, which is probably determined by the following cytoplasmic region. Interestingly, in contrast to HCMV gB in prefusion conformation, the resolved α-helix of the MPR in HSV-1 does not show any direct interaction with the fusion loops [118], which could indicate the release of the fusion loops from the MPR to be a pre-requisite for fusion triggering, similar to the required cleavage in class I fusion proteins.

## 8. Discussion

The structural determination of any viral membrane fusion protein is a challenge. The described methods to stabilise the metastable prefusion form shows the range of possibilities researchers have to stabilise their fusion protein of interest (Figure 2). Although many of the approaches were used to stabilise class I fusion proteins, there are common features found in all three classes that can be targeted, as seen with the disulfide bond linkage in class II proteins and the helix breaker insertion in class III (Table 1). New and powerful structure prediction programs such as Alpha fold [125] might soon be able to generate viral fusion protein structures in silico for which no structural data are available in order to successfully guide mutational approaches. If possible, this could become a very powerful and efficient strategy to create prefusion-stabilised fusion proteins of previously unresolved structures.

The recent prefusion structure determination of HCMV [41] and HSV-1 gB [43] marks an important step towards the utilisation of these proteins for immunogen development. The functional and structural stabilisation of HSV-1 gB via the introduction of a proline substitution in a conserved, structural element might make this an approach applicable to other class III fusion proteins, as illustrated by the successful proline substitution in the corresponding residue in VSV G [43]. Nevertheless, the fact that this stabilisation is only effective in the context of a membrane prompts the need for further stabilising mutations that could now be designed using the available homologous, high-resolution HCMV gB prefusion structure [41].

In order to fully understand the mode of action of a membrane fusion protein from a basic science perspective, structural information of the pre- and postfusion and also the intermediate states (Figure 1) is necessary. Since the latter conformations are often very short lived and unstable, they are hard to determine structurally. It will be exciting to see if structure prediction programs will be able to model these states at some point to allow stabilisation via intermediate state tailored mutations or specific inhibitors. Then, the challenge would be to verify the existence of these stabilised states in the fusion process of natural proteins. In addition, the membrane embedded regions of many viral fusion proteins remain structurally unresolved, despite their important roles in fusion control and execution as suggested for herpesvirus gB [41,118]. The continuous development of membrane mimetic systems such as nanodiscs, amphipols or peptidiscs [126] as well as the continuing advancements in cryoET [127] will facilitate the structural analysis of the full-length proteins in their native lipid environment in the future.

Taken together, a better understanding of the molecular mechanisms of the fusion process combined with the expanding toolkit described in this review and the utilisation of powerful structure prediction methods has the potential to create a framework to efficiently stabilise viral membrane fusion proteins in prefusion conformation. The medical relevance of the prefusion conformation is demonstrated by the successful use of stabilised fusogens for vaccine and antiviral drug development [61,80,128] illustrating the need to focus research efforts in this area to fight and be prepared against upcoming viral diseases.

## Figures and Tables

**Figure 1 viruses-14-01816-f001:**
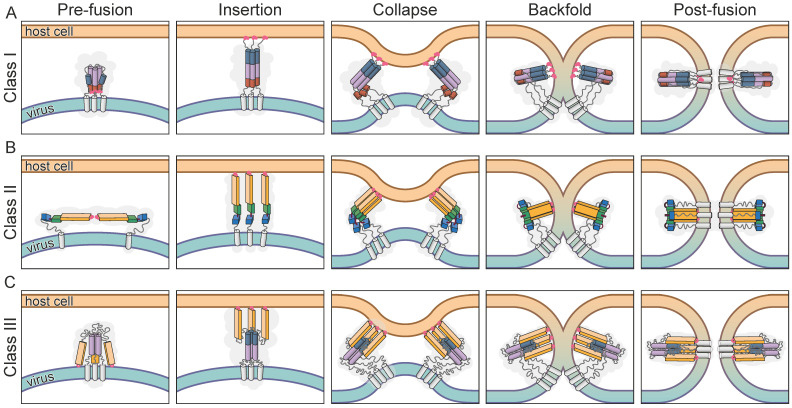
Three classes of viral membrane fusion proteins. Viral fusion class models illustrated by key domains in different states of the fusion process. The fusion proteins go through a number of conformations that drive the membrane fusion from approximation through hemifusion to fusion pore opening. Examples of structures are available for the pre- and postfusion states in all classes in contrast to intermediate conformations that are unstable and mostly extrapolated. Fusion loops or peptides are shown in pink, transmembrane domains as white cylinders while loop regions as well as regions not characteristic for the class are depicted as grey lines. (**A**) In class I fusion proteins, the long central coiled-coil domain (purple cylinders) is extended by refolding of the loop region connecting the short α-helical region (blue cylinder). After fusion loop anchoring the insertion state collapses, driven by the backfolding of the C-terminal portion of the helix (dark orange cylinder) to form a 6-helix bundle. (**B**) Class II fusogens rearrange after pH triggering, during insertion before the final trimer is formed. The fusion loops are located at the tip of a β-sheet in domain II (light orange box), while the motion is driven by a rotation of domain III (blue box) around domain I (green box) using a short hinge region (brown). (**C**) Class III is defined by an α-helical core similar to class I (purple cylinders), with a potential formation of a helical extension during insertion (blue cylinder), while the fusion loops are located at the tip of a β-sheet domain (light orange box) similar to class II.

**Table 1 viruses-14-01816-t001:** Overview of viral membrane fusion proteins stabilised in pre-fusion conformation, grouped by stabilisation method.

Class	Virus	Protein	Mutation(s)	Trimerisation	Cleavage Site Modification	Expression Construct	Fusion Activity	Design Method	PDB Number	Reference
**Proline substitution(s)**										
I	hMPV	F	A185P	foldon	ENVRRRR substitution	ectodomain	N.D.	design based on homologous structure (RSV)	5WB0	[57]
I	Influenza A	HA	V55P, S71P	-	-	full protein	impaired in red blood cell fusion assay	based on structure	-	[58]
I	MERS-CoV	S	V1060P, L1061P	foldon	ASVG substitution	ectodomain	impaired in pseudotyped lentivirus infection assays	design based on homologous structures (RSV, HIV, HCoV-HKU1)	5W9I	[59]
I	SARS-CoV	S	K968P, V969P	foldon	-	ectodomain	N.D.	based on structure	6CRZ	[60]
I	SARS-CoV-2	S	K968P, V969P, A942P, F817P, A892P, A899P	foldon	GSAS substitution	ectodomain	N.D.	based on structure	6XKL	[61]
I	SARS-CoV-2	S	K986P, V987P	foldon	GSAS substitution	ectodomain	N.D.	based on structure	6VSB	[62]
I	SARS-CoV-2	S	A892P, A942P, V987P	foldon	GSAS substitution	ectodomain	impaired in cell–cell fusion assay	based on structure	7A4N, 7AD1	[63]
III	HSV-1	gB	H516P	-	-	full protein	impaired in cell–cell fusion assay	molecular dynamics simulation	6Z9M	[43]
**Cystein-linkage strategy**										
I	HeV	F	Y97C + G131C, N100C + A119C	GCN4	-	ectodomain	impaired in cell–cell fusion assay	based on structure	-	[64]
I	MV	F	I452C + G460C	-	-	full protein	impaired in cell–cell fusion assay	design based on homologous structure (PIV5)	-	[65]
I	PIV5	F	T481C + T482C	-	-	full protein	impaired in cell–cell fusion assay	based on structure	-	[66]
I	RSV	F	A149C + Y458C or N183C + N428C	foldon	-	ectodomain	N.D.	based on structure	_	[67]
I	HIV	GP	A501C + T605C	-	LRLRLR substitution	ectodomain	N.D.	mutation screening	-	[68]
I	Influenza A	HA	HA1: 212C + 216C	-	-	ectodomain	reduced in red blood cell fusion assay	mutation screening	-	[69]
II	DENV	E	A259C or A257C *1	-	-	ectodomain	N.D.	based on structure	-	[70]
II	DENV2	E	A259C or A257C *1, L107C+A313C	-	-	ectodomain	impaired for double mutant in liposome flotation assay	based on structure	-	[71]
II	ZIKV	E	A264C	-	-	ectodomain	impaired in pseudotyped VSV infection assays	design based on homologous structure (DENV)	-	[70,72]
II	ZIKV	E	L107C, A264C, A319C, W101A	-	-	ectodomain	N.D., but disrupted fusion loop epitop	design based on homologous structure (DENV)	-	[73]
**Class**	**Virus**	**Protein**	**Mutation(s)**	**Trimerisation**	**Cleavage site** **modification**	**Expression** **construct**	**Fusion activity**	**Design method**	**PDB** **number**	**Reference**
**Charge repulsion reduction mutations**										
I	Influenza A	HA	H26W, K51I, E103I	-	-	ectodomain	impaired in cell–cell fusion assay	based on structure	-	[74]
**Combination of different stabilisation modalities**										
I	Ebola	GP	T577P, K588F	-	*2	ectodomain	N.D.	design based on homologous structures (RSV, HIV)	6VKM	[75]
I	HIV	GP	I559P, A501C + T605C	-	LRLRLR substitution	ectodomain	N.D.	mutation screening	-	[76]
I	HIV	GP	A204I, I573F, K588E, D589V, N651F, K655I, I535N	-	RRRRRR substitution	ectodomain	N.D.	based on structure	6CK9	[77]
I	LASV	GP	R207C + G360C, E329P	-	RRRR substitution	ectodomain	impaired in pseudotyped VSV infection assays	design based on homologous structures (RSV, HIV)	5VK2	[78]
I	NiV	F	S191P, L172F, L104C + I114C	GCN4	-	ectodomain	N.D.	based on structure	-	[79]
I	RSV	F	S155C + S290C, S190F, V207L	foldon	-	ectodomain	N.D.	based on structure	4MMQ–4MMV, 5K6C	[80]
I	RSV	F	S215P, N67I	foldon	GSGSGR linker	ectodomain	N.D.	based on structure	5C6B, 5C69	[47]
I	SARS-CoV	S	A892P, A942P, V987P, D614N, R682S, R685G	foldon	GSA substitution	ectodomain	impaired in cell–cell fusion assay	based on structure	7A4N, 7AD1	[63]
I	hMPV	F	L110C, T127C, A140C, A147C, N153C, A185P, L219K, V231I, N322C, T365C, N368H, E453Q, V463C	foldon	RRRR substitution	ectodomain	N.D.	design based on structure and homologous structure (RSV)	7SEM, 7SEJ	[81]
**Other stabilisation methods**										
I	Influenza A	HA	R329Q (cleavage site mutation)	-	R329Q	full-length	indirect *3	based on structure and biochem data	-	[82]
I	Influenza A	HA	multiple	GCN4	R329Q	miniHA *4	N.D.	based on structure	5CJQ, 5CJS	[83]
I	MHV	S	R717S (cleavage site mutation)	GCN4	R717S	ectodomain	N.D.	biochem data	3JCL	[84]
I	RSV	F	S155C, S290C, S190F, V207L, L512C, L513C, G519C, K520C, M526C, I527C, 533C, 534C, 540C, 541C (cystein Zipper)	foldon	-	ectodomain	N.D.	based on structure	-	[85]

*1 serotype dependent, *2 mucin-like domain deleted (313–465), *3 conformational change detected using protease accessibility, *4 short form of the ectodomain. N.D. not determined.

## Data Availability

Not applicable.

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
