# Peer review of "Stabilisation of Viral Membrane Fusion Proteins in Prefusion Conformation by Structure-Based Design for Structure Determination and Vaccine Development"

_viruses, 2022, doi:10.3390/v14081816_

Round 1
Reviewer 1 Report
The review summarizes the membrane fusion mechanism of all three classes of viruses very well. The metastable nature of viral fusion proteins makes structural determination and vaccine development very challenging. The author includes a variety of structure-based stabilization methodology to lock the fusion protein in the prefusion state, making it suitable for structural study and vaccine development. Understanding viral fusion protein, which is not only of essential function during infection but also very often a major target of immune system, would be critical for developing anti-viral therapeutics. This review provides a comprehensive and insightful view on this critical procedure. Here are some comments for the author.
11. Viral membrane fusion is usually a target of antibody and small molecule drug. Steps as mentioned in Figure 1, such as prefusion and backfold, are captured by neutralizing antibodies, and the whole cascade would be blocked. In the case of HIV, small molecules targeting CD4 receptor binding site or peptides blocking the formation of post-fusion structure have been applied to the treatment to infection clinically. Antibodies and small molecule/peptide drugs targeting viral membrane fusion would also be an important part of the medical relevance.
22. It would be clearer if a receptor is added to the membrane fusion of class I virus in Figure 1.
33. The low pH is required for membrane fusion of class 2 virus, which should be indicated Figure 1.
44. Metastability of fusion protein is critical for the fusion activity. As mentioned in the review stabilizing viral fusion protein, although make it easier to produce soluble viral protein for structural studies, could impair the fusion activity. From this point of view, stabilisation approaches may also compromise the immunogenicity when applied for vaccine development, which occurred during the HIV vaccine development. This limitation should also be covered in the review.
Reviewer 2 Report
The review was well received by this reviewer and should be a welcome addition to the literature. In light of the recent interest in the stabilized Coronavirus S- structure, and the eagerly anticipated RSV vaccine trial(s) it is expected that stabilizing viral fusion glycoproteins in their pre-fusion conformations will be en vogue for years to come. In general, the review is well written and fairly easy to follow. I much enjoyed the simple descriptions of the stabilizing changes followed by the more detailed information. My only real comments come from the focus of the review on HSV and CMV gB. The work would benefit from a more in-depth discussion of a lot of biologically relevant “fusion machinery and fusion regulators’- especially given the topic and the decades of work involving multiple groups. I strongly encourage discussion of virus and cell proteins that may be involved with stabilizing and regulating pre-fusion structures including some previous work identifying conformational changes. Beyond the potential use in an (antibody-focused) vaccine, the beauty and functional relevance of the stabilized structure is best understood in the context of its biology.
Some suggested starting points are given:
Dollery, S. J., Delboy, M. G., & Nicola, A. V. (2010). Low pH-induced conformational change in herpes simplex virus glycoprotein B. Journal of virology, 84(8), 3759–3766. https://doi.org/10.1128/JVI.02573-09
Atanasiu, D., Saw, W. T., Eisenberg, R. J., & Cohen, G. H. (2016). Regulation of Herpes Simplex Virus Glycoprotein-Induced Cascade of Events Governing Cell-Cell Fusion. Journal of virology, 90(23), 10535–10544. https://doi.org/10.1128/JVI.01501-16
Muggeridge MI. Glycoprotein B of herpes simplex virus 2 has more than one intracellular conformation and is altered by low pH. J Virol. 2012 Jun;86(12):6444-56. doi: 10.1128/JVI.06668-11. Epub 2012 Apr 18. PMID: 22514344; PMCID: PMC3393537.
J. Fontana, D. Atanasiu, W. T. Saw, J. R. Gallagher, R. G. Cox, J. C. Whitbeck, L. M. Brown, R. J. Eisenberg, G. H. Cohen, The fusion loops of the initial prefusion conformation of herpes simplex virus 1 fusion protein point toward the membrane. mBio 8, e01268-17 (2017)
Stampfer SD, Lou H, Cohen GH, Eisenberg RJ, Heldwein EE. Structural basis of local, pH-dependent conformational changes in glycoprotein B from herpes simplex virus type 1. J Virol. 2010 Dec;84(24):12924-33. doi: 10.1128/JVI.01750-10. Epub 2010 Oct 13. PMID: 20943984; PMCID: PMC3004323.
Dollery, S. J., Wright, C. C., Johnson, D. C., & Nicola, A. V. (2011). Low-pH-dependent changes in the conformation and oligomeric state of the prefusion form of herpes simplex virus glycoprotein B are separable from fusion activity. Journal of virology, 85(19), 9964–9973. https://doi.org/10.1128/JVI.05291-11
Kogure, A., Uehori, J., Arase, N., Shiratori, I., Tanaka, S., Kawaguchi, Y., Spear, P. G., Lanier, L. L., & Arase, H. (2008). PILRalpha is a herpes simplex virus-1 entry coreceptor that associates with glycoprotein B. Cell, 132(6), 935–944. https://doi.org/10.1016/j.cell.2008.01.043
Komala Sari, T., Gianopulos, K. A., Weed, D. J., Schneider, S. M., Pritchard, S. M., & Nicola, A. V. (2020). Herpes Simplex Virus Glycoprotein C Regulates Low-pH Entry. mSphere, 5(1), e00826-19. https://doi.org/10.1128/mSphere.00826-19
Round 2
Reviewer 2 Report
Dear Authors,
My enthusiasm stands. I think this will be a very useful review to people.
The added and altered sections look fine given the focus of the review. I do feel that it adds to talk a little more about the “prototype herpesvirus” in the herpesvirus special edition. There are no more reasonable comments from this reviewer. I did not detect any typos.
Thank you.